# Cardiometabolic disorders, inflammation and the incidence of non-alcoholic fatty liver disease: A longitudinal study comparing lean and non-lean individuals

Ehimen C. Aneni[1]*, Gul Jana Saeed[2], Marcio Sommer Bittencourt[3,4], Miguel Cainzos-Achirica[5,6], Chukwuemeka U. Osondu[7], Matthew Budoff[8,9], Edison R. Parise[10], Raul D. Santos[4,11], Khurram Nasir[5,6]

1 Section of Cardiovascular Medicine, Department of Internal Medicine, Yale University School of Medicine, New Haven, CT, United States of America, 2 Center for Sleep and Cardiovascular Outcomes Research, University of Pittsburgh, Pittsburgh, PA, United States of America, 3 Center for Clinical and Epidemiological Research, University Hospital and State of São Paulo Cancer Institute (ICESP), University of São Paulo, São Paulo, Brazil, 4 Hospital Israelita Albert Einstein, São Paulo, Brazil, 5 Division of Cardiovascular Prevention and Wellness, Houston Methodist DeBakey Heart and Vascular Center, Houston, TX, United States of America, 6 Center for Outcomes Research, Houston Methodist Research Institute, Houston, TX, United States of America, 7 Baptist Health South Florida, Miami, FL, United States of America, 8 The Lundquist Institute for Biomedical Innovation, Los Angeles, CA, United States of America, 9 David Geffen School of Medicine at UCLA, Los Angeles, CA, United States of America, 10 Federal University of Sao Paulo (UNIFESP), Sao Paulo, Brazil, 11 Lipid Clinic Heart Institute (InCor), University of São Paulo Medical School Hospital, São Paulo, Brazil

* ehianeni@outlook.com

**Data Availability Statement:** All relevant data are within the paper and its Supporting Information files.

## Abstract

### Background

There is limited knowledge about the risk of non-alcoholic fatty liver disease (NAFLD) associated with cardiometabolic disorders in lean persons. This study examines the contribution of cardiometabolic disorders to NAFLD risk among lean individuals and compares to non-lean individuals.

### Methods

We analyzed longitudinal data from 6,513 participants of a yearly voluntary routine health testing conducted at the Hospital Israelita Albert Einstein, Brazil. NAFLD was defined as hepatic ultrasound diagnosed fatty liver in individuals scoring below 8 on the alcohol use disorders identification test. Our main exposure variables were elevated blood glucose, elevated blood pressure (BP), presence of atherogenic dyslipidemia (AD, defined as the combination of elevated triglycerides and low HDL cholesterol) and physical inactivity (<150 minutes/week of moderate activity). We further assessed the risk of NAFLD with elevations in waist circumference and high sensitivity C-reactive protein (HsCRP).

**Funding:** RDS has received honoraria related to consulting, speaking, and research activities from Ache, Amgen, AstraZeneca, Esperion, Kowa, Merck, Novo-Nordisk, Pfizer, PTC, and Sanofi/Regeneron. MSB has received honoraria related to consulting, speaking and research activities from Boston Scientific, Sanofi, GE HealthCare, EMS and Novo-Nordisk. the funders had no role in study design, data collection and analysis, decision to publish, or preparation of the manuscript. The funders had no role in study design, data collection and analysis, decision to publish, or preparation of the manuscript.

**Competing interests:** The authors have declared that no competing interests exist.

## Results

Over 15,580 person-years (PY) of follow-up, the incidence rate of NAFLD was 7.7 per 100 PY. In multivariate analysis adjusting for likely confounders, AD was associated with a 72% greater risk of NAFLD (IRR: 1.72 [95% CI:1.32–2.23]). Elevated blood glucose (IRR: 1.71 [95%CI: 1.29–2.28]) and physical inactivity (IRR: 1.46 [95%CI: 1.28–1.66]) were also independently associated with increased risk of NAFLD. In lean individuals, AD, elevated blood glucose and elevated BP were significantly associated with NAFLD although for elevated blood glucose, statistical significance was lost after adjusting for possible confounders. Physical inactivity and elevations in HsCRP were not associated with the risk of NAFLD in lean individuals only. Among lean (and non-lean) individuals, there was an independent association between progressively increasing waist circumference and NAFLD.

## Conclusion

Cardiometabolic risk factors are independently associated with NAFLD. However, there are significant differences in the metabolic risk predictors of NAFLD between lean and non-lean individuals. Personalized cardiovascular disease risk stratification and appropriate preventive measures should be considered in both lean and non-lean individuals to prevent the development of NAFLD.

## Introduction

Non-alcoholic fatty liver disease (NAFLD) is a chronic liver condition characterized by lipid accumulation in the hepatic cells of patients who do not consume alcohol in excess or have other competing etiologies such as medication use or hepatic viral disease [1]. Globally, the prevalence of NAFLD is high, estimated at 24% [2] and is expected to continue to rise largely driven by the obesity epidemic [3]. There is a strong relationship between NAFLD and BMI. In a nationally representative cohort of US adults, individuals with a BMI $\geq 35 kg/m^2$ were more than 4 times as likely to have NAFLD compared to those with BMI $<25 kg/m^2$ even after adjusting for comorbidities and alcohol use [4].

While very prevalent among obese individuals, NAFLD is not restricted to obese persons. Analysis from the National Health and Nutrition Examination Survey (NHANES) shows that the prevalence of NAFLD among lean individuals in the US (BMI $<25 kg/m^2$) is about 7% [5]. Emerging data suggests that lean persons with NAFLD have a distinct metabolic phenotype that is characterized by relatively less metabolic derangement than obese individuals with NAFLD but greater cardiometabolic risk than lean individuals without NAFLD [5]. Several studies have also shown increased all-cause and cardiovascular disease (CVD) related mortality among lean individuals with NAFLD compared to lean individuals without NAFLD suggesting that lean NAFLD is not a benign condition [6].

Much of the published literature that has examined the relationship between cardiometabolic risk factors and NAFLD has been cross-sectional in design. Only a handful of studies have assessed the temporal relationship between cardiometabolic risk and incident NAFLD and these have largely been in hospital-based registries with diagnoses made by hepatic serologies only [7]. Population studies examining the relationship between cardiometabolic disorders and incident NAFLD in both lean and non-lean individuals are lacking.

Understanding the risk of NAFLD associated with cardiometabolic disorders and the role of BMI can be informative in designing interventions to prevent the development of NAFLD and its downstream consequences. This is particularly important in lean individuals who would otherwise not be targeted for aggressive interventions. Thus, we conducted this analysis to estimate the incidence of NAFLD and to determine if cardiometabolic risk factors were drivers of NAFLD, particularly in lean individuals.

## Methods

This is a longitudinal study analyzing data collected from participants undergoing routine health promotion evaluation at the Preventive Medicine Center of the Hospital Israelita Albert Einstein in São Paulo, Brazil from 2004 to 2016. Participation was voluntary. Because of the deidentified nature of the dataset the local institutional review board deemed this research to be exempt from human subjects research review and the study was approved in compliance with the guidelines outlined in the 1975 Declaration of Helsinki and with local guidelines.

### Data collection

The health evaluation was conducted year-round and participants had one visit per calendar year. At each visit, the study participants filled out questionnaires on demographic characteristics and social history including current and prior cigarette smoking history, and levels of physical activity as assessed by the short form of the international physical activity questionnaire (IPAQ-SF). A detailed medical history was obtained to determine presence of underlying comorbidities such as diabetes mellitus, hypertension and use of medications. As part of the evaluation, measures of adiposity such as body mass index (obtained from the height and weight) and abdominal circumference were assessed. Blood pressure (BP) was obtained as a mean of three resting measures, the first of which was measured after a 5-minute rest and in accordance with the American Heart Association guidelines [8]. Fasting blood samples were also obtained for lipid profile (total cholesterol, HDL cholesterol, LDL cholesterol and triglycerides), blood glucose and high sensitivity C-reactive protein (HsCRP).

Plasma lipids and glucose were measured by standardized automated laboratory tests using a VITROS platform (Johnson & Johnson Clinical Diagnostics, New Brunswick, New Jersey). Total cholesterol was measured by an enzymatic colorimetric method, HDL-cholesterol was measured by a precipitation method while LDL-cholesterol was calculated using the Friedewald formula (for triglyceride levels less than 400mg/dl). High-sensitivity C-reactive protein (HsCRP) levels were determined by immunonephelometry (Dade-Behring GmbH, Mannheim, Germany). All laboratory testing was performed at the Central Laboratory of the Hospital Israelita Albert Einstein.

At each visit, participants had hepatic ultrasound to assess for the presence of hepatic steatosis. This was done after a 6 hour fast using a Siemens ACUSONXP-10 ultrasound machine (Siemens AG, Mountain View, California). The ultrasound images were read by board certified radiologists who were unaware of the clinical or laboratory data of the participants. Hepatic steatosis was assessed as the presence of increased hepatic echogenicity (brightness) relative to the right renal parenchyma [9].

Participants also filled the alcohol use disorders identification test (AUDIT), a 10-item questionnaire with scores ranging from 0–4 for each question. The presence of an AUDIT score ≥8 is associated with habitual harmful alcohol drinking [10].

## Outcomes

The primary outcome was incident NAFLD, defined as hepatic steatosis in individuals whose AUDIT score was less than 8.

**Exposure and covariate definitions.**   The main exposure variables were elevated blood pressure (BP), elevated blood glucose, atherogenic dyslipidemia (AD), elevated HsCRP and physical inactivity. These were chosen to reflect cardiometabolic risk factors that were prevalent in the community and were potentially modifiable. Elevated BP was defined as a systolic blood pressure (SBP) of 130 mmHg or more, a diastolic blood pressure (DBP) of 80mmHg or more, the use of medications to treat hypertension, and/or a self-reported history of hypertension. Elevated blood glucose was defined as a fasting blood glucose of 100mg/dl or greater, a history of diabetes or use of glucose lowering medication. Atherogenic dyslipidemia (AD) was defined as a combination of elevated triglycerides (>150mg/dl) and low HDL-cholesterol levels (<40mg/dl for men and <50mg/dl for women). Physical inactivity was defined as persons not achieving up to 150 minutes/week of moderate activity on the IPAQ-SF questionnaire. Body mass index (BMI) was calculated as the weight in kilogram (kg) divided by the square of the height in meters (expressed as $kg/m^2$). Based on their BMI, participants were categorized as lean (BMI 18.5–24.9 $kg/m^2$) or non-lean (BMI 25 $kg/m^2$ and above). To assess the influence of waist circumference on NAFLD in lean individuals, we recategorized waist circumference (a continuous variable) into sub-stratum specific tertiles.

**Statistical analysis.**   Analyses were restricted to participants with complete data at baseline visit and a follow-up visit that was at least 6 months after the baseline visit. We excluded participants with BMI less than 18.5kg/$m^2$, AUDIT scores $\geq$ 8, follow-up less than 6 months and those with NAFLD at baseline.

All continuous data were assessed for normality. Means with standard deviations (SD) were used to describe normally distributed continuous variables, while medians with interquartile ranges (IQR) were computed for non-normally distributed continuous variables. For categorical variables, the frequencies (%) were computed across the groups and compared using chi-square test.

We analyzed the incidence of NAFLD as the number of new cases in the population at-risk per 100 person-years (PY) of follow-up. Participants contributed to person-time of risk as long as they had not developed NAFLD. Censoring occurred when participants developed their first diagnosis of NAFLD or at their last exam if they never had a diagnosis of NAFLD. We compared the incidence rates using the Mantel-Cox method computing the incidence rate ratios and 95% confidence intervals (95% CI) for the relationship between each of the demographic and cardiometabolic risk factors and incident NAFLD. We created 2 models: the first was unadjusted, the second was adjusted for age, sex, cigarette smoking, use of lipid lowering medication (all exposure groups), abdominal circumference (all exposure groups except physical inactivity), the presence of diabetes, use of diabetes lowering medication (all exposure groups except elevated blood glucose), a history of hypertension, use of BP lowering medication (all exposure groups except elevated BP) and lipid lowering medication (all exposure groups except AD). Univariate analysis to assess the association between age-group and NAFLD, and sex and NAFLD was also conducted for BMI sub-groups (lean and non-lean). We also conducted BMI subgroup analyses for each of the primary exposures (univariate and multivariate analysis). For each subgroup analyses, we conducted the Mantel-Haenszel test for homogeneity to assess for possible interactions.

To assess for the impact of residual adiposity conducted analyses on the risk of NAFLD among BMI stratum specific tertiles of waist circumference. All statistical analyses were carried out on Stata software version 16 [11].

## Results

### Baseline characteristics

Fig 1 details the participant flow and exclusions for this study. A total of 33,875 individuals had complete data of interest and at least one study visit (baseline). We sequentially excluded those with AUDIT scores greater than 8 (4500 participants) and BMI <18.5Kg/m$^2$ at baseline (230 participants) leaving 29,145 participants. Of these 9,318 had at least one follow-up visit. From this we excluded individuals with NAFLD at baseline as well as those with less than 6 months of follow-up. The final study population included a total of 6513 participants with a follow-up of 15,580 person-years (PY). The mean age at baseline was 44 ±9 years and 32% were female. Forty-nine percent of participants were non-lean (overweight [BMI: 25–29.9 kg/m2] or obese [BMI ≥30 Kg/m2]). Details of the baseline characteristics of participants can be found in Table 1.

### Demographics and covariates and incident NAFLD

The incidence rate of NAFLD was 8 per 100 PY. The incidence was higher among males compared to females, those aged 45 years or more, persons with a history of hypertension or use of

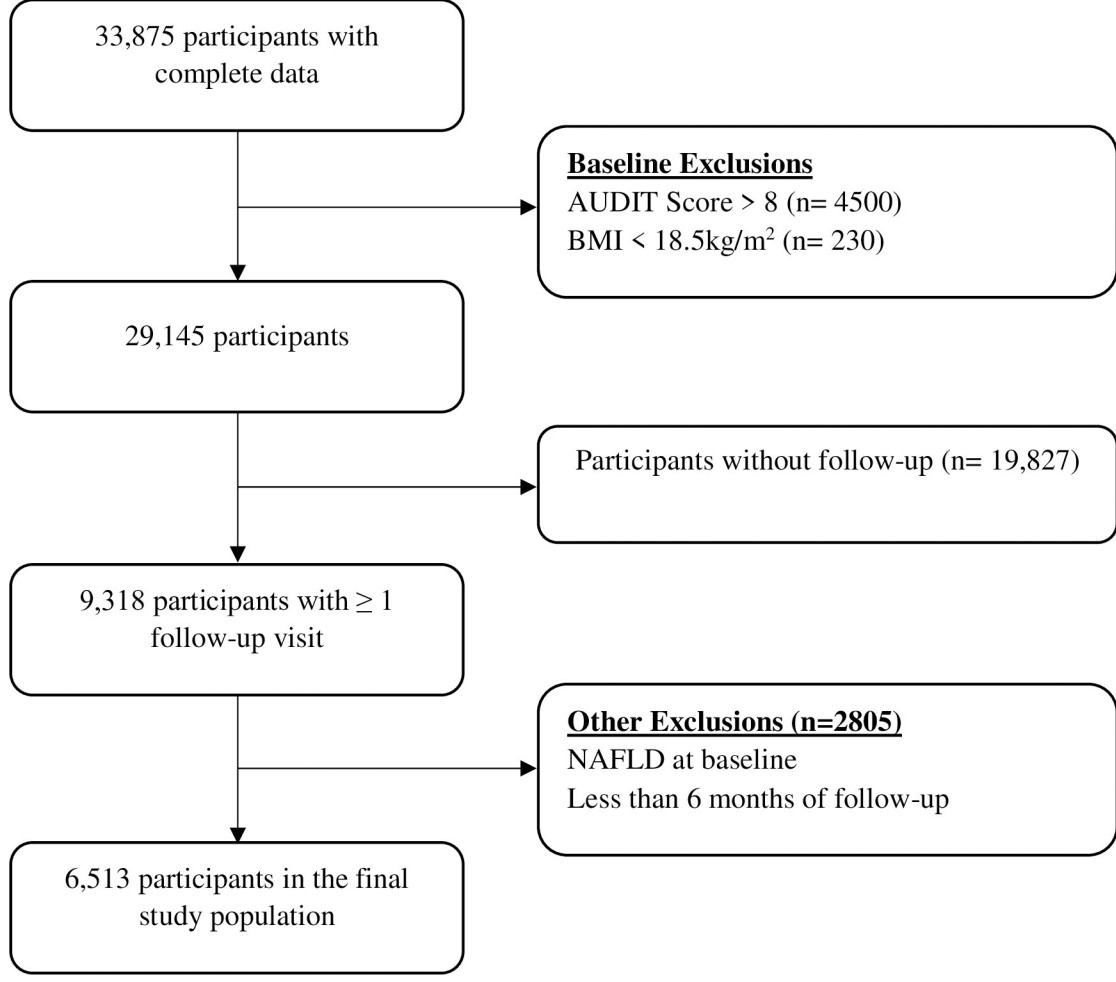

**Fig 1. Study participant flow and exclusions.** Participants with complete data at baseline (N = 33,875). Excluding participants with AUDIT score > 8 (n = 4500) and/or BMI < 18.5kg/m$^2$ (n = 230), N = 29145 were selected. Further excluding those without any follow-up visits (n = 19,827), those diagnosed with NAFLD at baseline or those with less than 6 months of follow-up data (n = 2805), the final number of study participants selected N = 6,513.

**Table 1. Characteristics of participants at baseline according to BMI categories.**

| Variable | All Participants | Lean (BMI: 18.5–24.9 Kg/m$^2$) | Overweight (BMI: 25–29.9 Kg/m$^2$) | Obese (BMI >30 Kg/m$^2$) | P value |
|---|---|---|---|---|---|
| N (%) | 6513 (100) | 3293 (50.6) | 2761 (42.4) | 459 (7.1) | |
| Mean age (y, SD) | 43.9 (8.7) | 42.6 (8.3) | 45.2 (8.8) | 45.6 (8.9) | <0.001 |
| Female Sex % | 32.4 | 44.2 | 19.7 | 23.3 | <0.001 |
| Mean BMI (Kg/m$^2$, SD) | 25.2 (3.2) | 22.7 (1.6) | 26.9 (1.3) | 32.4 (2.4) | <0.001 |
| Mean Waist Circumference (cm, SD) | 88.2 (10.8) | 81.4 (7.9) | 93.4 (7.0) | 106.1 (9.2) | <0.001 |
| Elevated Blood Pressure % | 43.1 | 29.2 | 55.5 | 68.7 | <0.001 |
| Elevated Blood Glucose % | 14.1 | 9.1 | 17.2 | 31.4 | <0.001 |
| Physical Activity <150minutes/ week % | 44.2 | 41.6 | 45.1 | 57.3 | <0.011 |
| Use of blood pressure lowering medication (%) | 9.9 | 5.6 | 13.3 | 20.3 | <0.001 |
| Use of lipid lowering medication (%) | 13.3 | 8.7 | 18.0 | 17.4 | <0.001 |
| Use of glucose lowering medication (%) | 2.4 | 1.4 | 2.8 | 7.4 | <0.001 |
| Smoking% | 6.1 | 5.7 | 6.5 | 7.2 | 0.257 |
| Atherogenic dyslipidemia (%) | 9.4 | 4.5 | 13.1 | 22.2 | 0.001 |
| Metabolic Syndrome (IDF) % | 2.2 | 0.2 | 3.2 | 9.8 | <0.001 |
| Median Total Cholesterol (mg/dL; IQR) | 188 (166–212) | 185 (164–208) | 192 (169–217) | 193 (167–219) | <0.001 |
| Median Triglycerides (mg/dL; IQR) | 95 (71–133) | 86 (64–115) | 107 (79–149) | 130 (94–180) | <0.001 |
| Median LDL-c (mg/dL; IQR) | 116 (95–138) | 111 (92–132) | 120 (99–142) | 119 (94–113) | <0.001 |
| Median HDL-c (mg/dL; IQR) | 49 (41–59) | 53 (44–63) | 46 (40–54) | 42 (37–50) | <0.001 |
| Median Fasting Glucose (mg/dL; IQR) | 85 (81–90) | 84 (79–88) | 87 (82–92) | 89 (83–95) | <0.001 |
| Median HsCRP (mg/dl, IQR) | 1.0 (0.5–2.1) | 0.8 (0.5–1.7) | 1.1 (0.6–2.2) | 2.2 (1.2–3.9) | <0.001 |
| Median AST (mg/dl, IQR) | 26 (22–31) | 24 (21–29) | 27 (23–31) | 27 (23–32) | <0.001 |
| Median ALT (mg/dl, IQR) | 31 (25–40) | 28 (23–36) | 34 (27–43) | 37 (29–46) | <0.001 |
| Median Alkaline Phosphatase (mg/dl, IQR) | 59 (50–69) | 57 (48–67) | 60 (52–71) | 65 (54–77) | <0.001 |

BP lowering medication, history of diabetes or use of glucose lowering medications and those with a history of hyperlipidemia. The incidence of NAFLD per 100PY among lean individuals was 3 compared to 11 and 23 among overweight and obese individuals respectively. More details can be found in Table 2.

## Atherogenic dyslipidemia

Persons with AD had nearly 3 times the risk of NAFLD in univariate analysis which remained statistically significant even after controlling for likely confounders (IRR: 1.94 [95% CI: 1.59–2.37], Table 5). When stratified by lean and non-lean groups, lean individuals had lower absolute incidence rates of NAFLD however the unadjusted and adjusted relative risk (rate ratio) was greater when comparing persons with AD to those without AD within the lean group (adjusted IRR 2.15 [95% CI: 1.02–4.53]) compared to non-lean groups (adjusted IRR 1.55 [95% CI: 1.36–1.90], P for interaction not significant). See Tables 3 and 4.

## Elevated blood glucose

The presence of elevated blood glucose was associated with about 70% increase in the risk of NAFLD (IRR: 1.71 [95% CI:1.29–2.28]) after controlling for likely confounders (Table 3). The incidence of NAFLD among lean individuals with elevated blood glucose was lower than non-lean individuals with elevated blood glucose. In unadjusted analysis, elevated blood glucose was associated with NAFLD in both lean and non- lean individuals however the statistical

**Table 2. Incidence rates and incidence rate ratios of NAFLD for demographic and cardiometabolic risk factors.**

| Variable | Incidence of NAFLD (per 100 PY) | Unadjusted Incidence Rate Ratio (95% CI) |
|---|---|---|
| Age 45 years and older | 10.3 (9.6–11.1) | 1.79 (1.60–2.01) |
| Age Less than 45 years (ref) | 5.8 (5.3–6.3) | |
| Male | 9.7 (9.1–10.3) | 2.62 (2.24–3.06) |
| Female (ref) | 3.7 (3.2–4.3) | |
| Smokers | 8.4 (6.8–10.4) | 1.06 (0.85–1.32) |
| Non-smokers (ref) | 7.7 (7.2–8.1) | |
| Elevated Waist Circumference | 18.0 (16.3–19.9) | 2.21 (2.58–3.29) |
| Normal Waist Circumference (ref) | 6.0 (5.6–6.5) | |
| Elevated total cholesterol (> = 200 mg/dL) | 9.2 (8.4–10.0) | 1.33 (1.19–1.49) |
| Normal total cholesterol (<200 mg/dL, ref) | 6.9 (6.4–7.4) | |
| History of hyperlipidemia | 11.0 (10.3–11.8) | 2.38 (2.11–2.69) |
| No known hyperlipidemia (ref) | 4.7 (4.2–5.2) | |
| Low HDL-c (<40 mg/dL in males and <50 mg/dL in females) | 12.2 (11.1–13.3) | 1.96 (1.74–2.20) |
| Normal HDL-c (ref) | 6.3 (5.9–6.8) | |
| Elevated LDL-c (> = 130 mg/dL) | 10.7 (9.2–12.5) | 1.42 (1.21–1.67) |
| Normal LDL-c (< 130 mg/dL, ref) | 7.4(7.0–7.9) | |
| Elevated triglycerides (> = 150 mg/dL) | 15.9 (14.5–17.4) | 2.67 (2.39–3.02) |
| Normal triglycerides (<150 mg/dL, ref) | 5.9 (5.5–6.3) | |
| History of Diabetes and / or Use of Glucose Lowering medication | 17.3 (7.1–8.0) | 2.31 (1.80–2.95) |
| No history of diabetes and no use of glucose lowering medication (ref) | 7.5 (13.6–22.0) | |
| History of hypertension or use of BP lowering medication | 13.2 (11.6–15.1) | 1.90 (1.64–2.20) |
| No history of hypertension AND no use of BP lowering medication (ref) | 7.1 (6.7–7.5) | |
| Glucose > = 100 mg/dL | 20.5 (17.6–23.8) | 2.95 (2.51–3.47) |
| Glucose <100 mg/dL (ref) | 7.0 (6.6–7.5) | |
| BMI 30 kg/m$^2$ or more (obese) | 23.2 (20.4–26.2) | 6.78 (5.90–7.80) |
| BMI 25–29.9 kg/m$^2$ (overweight) | 11.0 (10.3–11.8) | 3.39 (2.98–3.84) |
| BMI 18.5–24.9 kg/m$^2$ (lean, ref) | 2.9 (2.6–3.3) | ref |
| <150minutes/ week | 8.8 (6.3–7.4) | 1.28 (1.14–1.44) |
| $\geq$ 150minutes/ week (ref) | 6.8 (6.3–9.5) | ref |
| HsCRP $\geq$2 mg/dl | 10.4 (9.4–11.5) | 1.55 (1.37–1.74) |
| HsCRP <2 mg/dl (ref) | 6.8 (6.3–7.3) | ref |

Elevated Waist Circumference: Waist circumference >88cm in women or 102cm in men

LDL-c, low density lipoprotein cholesterol; HDL-c high density lipoprotein cholesterol; SD, standard deviation; IQR, interquartile range; HsCRP, high-sensitivity C-reactive protein; mg/dl, milligram per deciliter; BP, blood pressure

significance among the lean individuals was lost after adjusting for likely confounders (IRR: 1.45 95% CI: [0.75–2.79]). There was no significant interaction. See Table 4.

## Elevated blood pressure

In univariate analysis, elevated BP was associated with greater than 2-fold risk in NAFLD but lost its significance after controlling for likely confounders (IRR: 1.19 [95%: 0.96–1.48]). Lean

**Table 3. Cardiometabolic disorders and the risk of NAFLD.**

|  | Variable | Incidence of NAFLD (per 100 PY) | Unadjusted IRR (95% CI) | Adjusted IRR (95% CI) |
|---|---|---|---|---|
| Physical Inactivity[1] | <150minutes/ week | 8.8 (8.1–9.4) | 1.25 (1.12–1.40) | 1.46 (1.28–1.66) |
|  | ≥ 150minutes/ week (ref) | 6.9 (6.3–7.4) |  |  |
| Atherogenic Dyslipidemia[2] | Present | 18.3 (16.2–20.6) | 2.75 (2.40–3.15) | 1.72 (1.32–2.23) |
|  | Absent (ref) | 6.7 (6.3–7.1) |  |  |
| Blood pressure[3] | Elevated Blood Pressure | 11.2 (10.4–12.0) | 2.15 (1.91–2.41) | 1.19 (0.96–1.48) |
|  | BP not elevated (ref) | 5.2 (4.7–5.7) |  |  |
| Glucose[4] | Elevated Glucose | 19.0 (16.6–21.8) | 2.77 (2.39–3.22) | 1.71 (1.29–2.28) |
|  | Glucose normal (ref) | 6.9 (6.5–7.4) |  |  |
| HsCRP[1] | Elevated | 10.4 (9.4–11.5) | 1.54 (1.37–1.74) | 1.35 (1.12–1.63) |
|  | Normal | 6.8 (6.3–7.3) |  |  |

1. Adjusted for age, sex, cigarette smoking at baseline, history of hypertension, history of diabetes, use of lipid lowering medication, use of glucose lowering medication, use of blood pressure lowering medication

2. Adjusted for age, sex, cigarette smoking, history of hypertension, history of diabetes, use of glucose lowering medication, use of blood pressure lowering medication and waist circumference

3. Adjusted for age, sex, cigarette smoking, history of diabetes, use of glucose lowering medication, use of lipid lowering medication and waist circumference

4. Adjusted for age, sex, cigarette smoking, history of hypertension, use of blood pressure lowering medication, use of lipid lowering medication and waist circumference

Atherogenic Dyslipidemia: Defined as a combination of elevated triglycerides (≥150 mg/dL) AND low HDL-c (<40 mg/dL in men or <50 mg/dL in women).

**Table 4. Cardiometabolic disorders and the risk of NAFLD in lean and non-lean participants.**

|  |  | BMI <25 kg/m² | | | BMI ≥25 kg/m² | | |
|---|---|---|---|---|---|---|---|
|  | Variable | Incidence (per 100 PY) | Unadjusted IRR (95% CI) | Adjusted IRR (95% CI) | Incidence (per 100 PY) | Unadjusted IRR (95% CI) | Adjusted IRR (95% CI) |
| Physical Activity[1] | <150minutes/ week | 3.2 | 1.14 (0.881–1.48) | 1.09 (0.80–1.50) | 14.0 | 1.20 (1.06–1.36) | 1.32 (1.14–1.54) |
|  | ≥ 150minutes/ week (ref) | 2.7 |  |  | 11.4 |  |  |
| Atherogenic Dyslipidemia[2] | Present | 8.8 | 3.39 (3.00–5.01) | 2.15 (1.02–4.53) | 21.1 | 1.86 (1.60–2.15) | 1.55 (1.15–2.09) |
|  | Absent (ref) | 2.7 |  |  | 11.2 |  |  |
| Blood Pressure[3*] | Elevated Blood Pressure | 5.1 | 2.46 (1.90–3.18) | 1.83 (1.08–3.10) | 14.2 | 1.32 (1.16–1.51) | 1.01 (0.78–1.31) |
|  | BP not elevated (ref) | 2.1 |  |  | 10.6 |  |  |
| Glucose[4] | Elevated Glucose | 7.9 | 2.84 (1.85–4.37) | 1.45 (0.75–2.79) | 23.1 | 2.03 (1.73–2.36) | 1.66 (1.17–12.37) |
|  | Glucose normal (ref) | 2.7 |  |  | 11.5 |  |  |
| HsCRP[1] | HsCRP > = 2 | 2.9 | 1.01 (0.74–1.39) | 1.11 (0.72–1.72) | 15.6 | 1.37 (1.20–1.56) | 1.31 (1.06–1.63) |
|  | HsCRP <2 | 2.9 |  |  | 11.4 |  |  |

1. Adjusted for age, sex, cigarette smoking at baseline, history of hypertension, history of diabetes, use of lipid lowering medication, use of glucose lowering medication, use of blood pressure lowering medication

2. Adjusted for age, sex, cigarette smoking, history of hypertension, history of diabetes, use of glucose lowering medication, use of blood pressure lowering medication and waist circumference

3. Adjusted for age, sex, cigarette smoking, history of diabetes, use of glucose lowering medication, use of lipid lowering medication and waist circumference

4. Adjusted for age, sex, cigarette smoking, history of hypertension, use of blood pressure lowering medication, use of lipid lowering medication and waist circumference

Atherogenic Dyslipidemia: Defined as a combination of elevated triglycerides (≥150 mg/dL) AND low HDL-c (<40 mg/dL in men or <50 mg/dL in women).

*Mantel-Haenszel test of unequal risk ratios was significant for only blood pressure at the p<0.05 level in both the unadjusted and adjusted analysis.

**Table 5. Baseline waist circumference and the risk of NAFLD in lean and non-lean individuals.**

| | BMI <25 kg/m$^2$ | | | BMI ≥25 kg/m$^2$ | | |
|---|---|---|---|---|---|---|
| Tertiles of Waist Circumference (N) | T1 (1155) | T2 (1232) | T3 (917) | T1 (1566) | T2 (1083) | T3 (574) |
| Baseline Mean Waist Circumference (± SD, kg/m$^2$) | 72.9 ± 4.9 | 82.6 ± 2.3 | 90.5 ± 3.2 | 88.5 ± 5.3 | 98.1 ± 2.2 | 108 ± 5.3 |
| Baseline Mean BMI (± SD, kg/m$^2$) | 21.5 ± 1.5 | 23.1 ± 1.2 | 23.8 ± 0.8 | 26.5 ± 1.4 | 27.7 ± 1.8 | 30.8 ± 3.0 |
| Incidence Rate per 100 PY (CI) | 0.8 (0.6–1.3) | 2.6 (2.1–3.3) | 6.0 (5.0–7.0) | 7.7 (6.9–8.7) | 14.8 (13.4–16.5) | 21.7 (19.4–24.2) |
| Unadjusted Rate Ratio | ref | 3.1 (1.9–4.9) | 7.1 (4.5–11.0) | ref | 1.9 (1.6–2.2) | 2.9 (2.4–3.3) |
| Adjusted* Rate Ratio | ref | 1.68 (0.91–3.10) | 4.62 (2.08–10.26) | ref | 1.79 (1.51–2.13) | 2.58 (2.14–3.11) |

individuals had much lower incidence of NAFLD regardless of their blood pressure status. Among lean individuals, the risk of NAFLD was 60% (IRR: 1.83 [95% CI:1.08–3.10]) greater among those with elevated BP compared to those without elevated BP, even after controlling for confounders. However, among non-lean individuals, although there was a univariate association between elevated BP and incident NAFLD, this was no longer statistically significant after controlling for likely confounders (IRR: 1.01 [95% CI: 0.78–1.31]). There was significant interaction between BMI group and elevated blood pressure on the risk of NAFLD (adjusted *P* for interaction = 0.012). See Tables 3 and 4.

### Physical inactivity

In univariate analysis persons who were categorized as physically inactive were 25% more likely to develop NAFLD. This association remained statistically significant after controlling for likely confounders (IRR: 1.46 [95% CI: 1.28–1.66]). In lean individuals, there was no association between levels of physical activity and the risk of NAFLD. However, in non-lean individuals, decreased physical activity was associated with incident NAFLD in both univariate and multivariate analysis (adjusted IRR 1.32 [95% CI: 1.14–1.54]) (See Table 4).

### HsCRP

Elevated CRP was associated with a greater than 50% in the risk of NAFLD. In multivariate analysis, this association persisted although slightly attenuated (IRR: 1.35 [95% CI: 1.12–1.63]). The incidence of elevated CRP was much lower among lean individuals (2.9 per 100PY) compared to non-lean individuals (15.6 per 100PY). While, elevated HsCRP was not associated with development of NAFLD in lean individuals, it was associated with a 30% increase in the risk of NAFLD in non-lean individuals (IRR: 1.32 [95% CI: 1.06–1.63]).

### Waist circumference, bmi and incident NAFLD

In lean individuals, the BMI increased with increasing tertiles of waist circumference. Among the lean, the incidence of NAFLD increased from 0.8 per 100PY in the lowest tertile to 6.0 per 100 PY in the highest tertile. There was also similar increase across tertiles of waist circumference in the non-lean individuals. Adjusting for possible confounders, persons in the highest tertile of waist circumference in both BMI categories had higher risk of NAFLD compared to the lowest tertile. Among the lean, this corresponded to greater than 4 times the risk (IRR 4.62 [95% CI: 2.08–10.26]). Details of these analyses can be found in Table 5.

### Discussion

In this longitudinal study of relatively young and asymptomatic individuals, cardiometabolic risk factors such as physical inactivity, elevated glucose and atherogenic dyslipidemia were

independently associated with incident NAFLD. In addition, among lean individuals, NAFLD risk was predicted by cardiometabolic risk factors. Physical inactivity and elevated HsCRP were not risk factors for incident NAFLD in lean individuals.

In our study, the incidence of NAFLD was approximately 8 per 100PY. This incidence is similar to the pooled incidence from Asian studies of 5.2 per 100 PY [12, 13]. In a study from Olmstead county in the US, the annual incidence of NAFLD in 2014 was 0.3 per 100 PY. The substantially lower incidence in the Olmstead county study may be due to differences in ethnic distribution (largely Caucasian / white, non-Hispanic) and methods of ascertaining NAFLD (NAFLD was identified by ICD codes, compared to liver ultrasound measurements in our study). We found no other study from a similar geographic population i.e. Brazilian or South American to compare to our study. In our study, obese persons had nearly 7-fold incidence rates compared to lean individuals confirming findings from other studies suggesting that obesity is a major driver of NAFLD incidence [14, 15].

There is limited information regarding the incidence of NAFLD in lean individuals. In our population, the incidence of NAFLD among lean individuals was relatively low, estimated at approximately 3 per 100PY. In a recent meta-analysis involving 9132 participants, the incidence of NAFLD among lean and non-obese participants was similar to our population (2.4 per 100PY) [16].

Although there have been several cross-sectional studies assessing the odds of NAFLD among persons with cardiometabolic risk factors, longitudinal population-based studies estimating this risk are scant. In the Framingham cohort, systolic and diastolic BP, triglyceride and fasting glucose levels were independent risk factors for incident hepatic steatosis [17]. Similarly, in a community-based population study conducted among 778 individuals free from NAFLD in Sri-Lanka, elevated BMI, diabetes and elevated triglycerides were independently associated with incident NAFLD [18]. Our study confirms these findings by showing independent relationship between elevations in blood glucose, the presence of atherogenic dyslipidemia and incident NAFLD.

Little is known about the association of cardiometabolic risk factors and incident NAFLD among lean individuals. In a recent study of 294 lean individuals without NAFLD who were followed for 7 years, the annual NAFLD incidence of 4.1% was comparable to our findings of about 3 per 100PY. In the same study, the presence of diabetes at baseline was the only independent cardiometabolic risk factor that was predictive of developing NAFLD [19]. The present study, which has significantly higher sample size (over 3000 lean participants), shows that elevated BP and the presence of atherogenic dyslipidemia were independent risk factors for incident NAFLD among lean individuals. Blood glucose was not an independent risk factor for incident NAFLD in lean individuals.

Interestingly, the independent relationship between elevated BP and incident NAFLD seen in lean persons was not seen in non-lean individuals. It is not readily clear what is responsible for this interaction; however, it may indicate that elevated BP is a stronger marker for insulin resistance among the lean compared to the non-lean. These findings raise more questions about the mechanistic differences in the development of NAFLD between lean and non-lean individuals and warrant further scientific investigation.

Increased physical activity has been thought to be protective of NAFLD [20]. In our study, physical inactivity was associated with greater risk of NAFLD. The benefits of physical activity on NAFLD risk have been previously assessed in a subset of this population. In that study, among those who were free from hepatic steatosis at baseline, people who were active at baseline (i.e. ≥150 minutes per week) and remained physically active at follow-up or moved from being inactive at baseline to being active at follow-up were less likely to develop hepatic steatosis compared to those who were inactive at baseline and follow-up [21]. In the present study,

there was no association between lack of physical activity and incidence of NAFLD among lean individuals. The reason for this observation is not readily clear from our data. However, it suggests that lack of physical activity in the absence of significant weight gain may not be a risk factor for NAFLD. This finding is worth further exploration in appropriately designed longitudinal/interventional studies.

Elevated HsCRP, a marker of inflammation, was associated with NAFLD in the entire population but not among the lean. Previous studies have shown a cross-sectional association between elevations in HsCRP and NAFLD [22]. However, only a few prospective studies have examined the relationship between HsCRP and incident NAFLD. In a recent study of over 4000 men, Lee and colleagues demonstrated that the risk of NAFLD increased with higher HsCRP values albeit within normal range thus supporting our earlier stated findings. To our knowledge, our study is the first to examine the risk of NAFLD with low grade inflammation in lean individuals. The absence of an association suggests that the low-grade inflammation is likely driven by adiposity which in turn is a greater determinant of NAFLD.

In the present study, we observed that higher waist circumferences in individuals who were lean (BMI $<25 kg/m^2$) was associated with increased risk of NAFLD. Lean individuals in the lowest tertile of waist circumference had very low incidence of NAFLD (<1 per 100PY, Table 5). This finding suggests that residual unmeasured adiposity in lean individuals may partly account for the increased risk of NAFLD. Studies have suggested that the pathophysiology of NAFLD in lean individuals may be related to more dysfunctional fat such as visceral obesity thus supporting our earlier stated assertion [23].

Taken together, our study findings highlight the impact of cardiometabolic disease on the incidence of NAFLD, regardless of the BMI profile of the patient. Particularly, it highlights the significance of cardiometabolic risk on the development of NAFLD in lean individuals. The consequences of NAFLD can be severe. If unchecked, NAFLD can progress to non-alcoholic steatohepatitis (NASH) and this in turn can lead to severe liver disease such as hepatic cirrhosis and hepatocellular cancer [24, 25]. NAFLD is also associated with cardiovascular disorders such as myocardial infarction and arrhythmias [26–28]. In addition to being associated with NAFLD, the presence of cardiometabolic disease, particularly diabetes, predicts NAFLD mortality [29].

Until recently, the focus of NAFLD had been largely among obese populations; however, emerging data now suggests that lean NAFLD presents a unique clinical phenotype and may not be a metabolically benign condition [16, 30]. Our study suggests that although lean individuals have much lower incidences of NAFLD, the presence of cardiometabolic risk factors including unmeasured adiposity plays a major role in the development of NAFLD in lean individuals. The downstream effects of NAFLD in lean individuals, particularly hard outcomes like mortality, and adverse cardiovascular and hepatic events require further exploration.

Given the known risk of adverse cardiovascular and hepatic outcomes related to NAFLD, including the lean subtype, there is a dire public health need to prevent NAFLD [6]. This study suggests additional measures of adiposity such as waist circumference, as well as easily implementable clinic measures such as BP measurement, serum measures of glucose and lipids which help identify individuals who are at risk for NAFLD particularly in lean individuals (BMI $<25Kg/m^2$) who are, otherwise, less likely to be assessed for NAFLD. Future studies should investigate population health systems that can easily identify these disorders and trigger aggressive preventive measures in order to reduce the risk of NAFLD.

The main strengths of this study are the longitudinal nature of the data which allows for assessment of temporality and the large population sample making it possible to assess NAFLD risk across several subgroups. To our knowledge this is the first study to examine

cardiometabolic disorders and inflammation related risk of incident NAFLD with emphasis in lean populations.

The study is limited by its retrospective design in which several variables such as medication groups, details on other liver conditions such as viral hepatitis, biomarkers of NAFLD severity (such as cytokeratin 18 fragment, interleukins, tissue necrosis factor) and genetic data were not collected. No direct markers of insulin resistance were measured; however, indirect measures such as atherogenic dyslipidemia (the combination of elevated triglyceride and low HDL-c) were computed for this analysis. Hepatic steatosis was diagnosed using ultrasound which is limited in its sensitivity [31]. However, it is a safe, practical and noninvasive method for assessing and reassessing the presence of hepatic steatosis [32]. Although effort was made to identify potential confounders and adjust for them in our analysis, there is still a possibility of residual confounding. Approximately 68% of participants did not have follow-up data and this may have introduced selection bias. Lastly, our study was conducted among relatively younger, largely male, Brazilians and may not be generalizable to populations outside of those demographics.

## Conclusions

In this large cohort of relatively young working-class Brazilians, cardiometabolic disorders and the presence of low-grade inflammation were associated with incident NAFLD. Lean individuals who eventually developed NAFLD exhibited a slightly different phenotype and residual adiposity may be a major driver of NAFLD incidence in lean people. Our findings suggest that interventions targeted at reducing cardiometabolic risk will also reduce the risk of NAFLD. These data also highlight the need to elucidate and aggressively modify cardiometabolic risk in lean individuals, who would have otherwise been perceived as low risk.

## Supporting information

**S1 Checklist. STROBE (Strengthening The Reporting of OBservational Studies in Epidemiology) checklist.**
(PDF)

**S1 Data.**
(XLSX)

**S2 Data.**
(XLS)

## Author Contributions

**Conceptualization:** Ehimen C. Aneni, Matthew Budoff, Khurram Nasir.

**Data curation:** Ehimen C. Aneni, Chukwuemeka U. Osondu.

**Formal analysis:** Ehimen C. Aneni, Chukwuemeka U. Osondu.

**Investigation:** Ehimen C. Aneni.

**Methodology:** Ehimen C. Aneni, Chukwuemeka U. Osondu, Raul D. Santos.

**Project administration:** Raul D. Santos.

**Software:** Chukwuemeka U. Osondu.

**Supervision:** Ehimen C. Aneni, Marcio Sommer Bittencourt, Raul D. Santos, Khurram Nasir.

**Validation:** Marcio Sommer Bittencourt, Chukwuemeka U. Osondu, Edison R. Parise, Raul D. Santos.

**Writing – original draft:** Ehimen C. Aneni, Gul Jana Saeed.

**Writing – review & editing:** Ehimen C. Aneni, Gul Jana Saeed, Marcio Sommer Bittencourt, Miguel Cainzos-Achirica, Chukwuemeka U. Osondu, Matthew Budoff, Edison R. Parise, Raul D. Santos, Khurram Nasir.

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
