## [Decision Letter · Decision Letter 0]

2 Aug 2021

PONE-D-21-09198

Cardiometabolic Disorders, Inflammation and the Incidence of Non-Alcoholic Fatty Liver Disease: A Longitudinal Study Comparing Lean and Non-lean Individuals

PLOS ONE

Dear Dr. Aneni,

Thank you for submitting your manuscript to PLOS ONE. After careful consideration, we feel that it has merit but does not fully meet PLOS ONE’s publication criteria as it currently stands. Therefore, we invite you to submit a revised version of the manuscript that addresses the points raised during the review process.

We look forward to receiving your revised manuscript.

Kind regards,

Martin Senechal, PhD

Academic Editor

PLOS ONE

Journal Requirements:

2. Please include additional information regarding the survey or questionnaire used in the study and ensure that you have provided sufficient details that others could replicate the analyses. For instance, if you developed a questionnaire as part of this study and it is not under a copyright more restrictive than CC-BY, please include a copy, in both the original language and English, as Supporting Information.  If the original language is written in non-Latin characters, for example Amharic, Chinese, or Korean, please use a file format that ensures these characters are visible.

Reviewers' comments:

Reviewer's Responses to Questions

**Comments to the Author**

1. Is the manuscript technically sound, and do the data support the conclusions?

Reviewer #1: Yes

2. Has the statistical analysis been performed appropriately and rigorously? 

Reviewer #1: Yes

3. Have the authors made all data underlying the findings in their manuscript fully available?

Reviewer #1: No

4. Is the manuscript presented in an intelligible fashion and written in standard English?

Reviewer #1: Yes

5. Review Comments to the Author

**Editor Comments:**

-What is the rationale for choosing 150 METs/min /week?

-METs and minutes per week are used interchangeably. Since they are not the same, please clarify what has been used. Is it 150 minutes/week of PA or 150 METs per week or 150METs/min/week. This need clarification.

-Why the author excluded participants with a BMI under 18 (kg/m2). Is there a rationale for NAFLD?

-Did the author kept only people with Data for NAFLD at baseline AND follow up? This needs to be clarified as well as how many people lost for these variables.

-I encourage the author to look at the checklist of the STROBE and address each point. For example, inclusion criteria and hypothesis should be in the manuscript or more precise. This is important since PLOS One Criteria require each study to be reported according to the appropriate guidelines.

Reviewer #1: The current study evaluates cardiometabolic risk factors among lean and non-lean individuals who were followed for the development of NAFLD. The study showed some significant differences between cardiometabolic profile of lean and non-lean individuals that developed NAFLD during follow up. Few comments:

1. It seems like participants were followed every 6months to one year. It is not clear why patients underwent hepatic USG at every visit? Was it based on clinical indication or specific criterias or all the participants underwent as part of screening protocol

2. Abstract: “In lean individuals, AD, elevated blood glucose and elevated BP were significantly associated with NAFLD”. Table 4 should elevated glucose was not associated with NAFLD among lean individuals in adjusted model whereas elevated BP and AD were.

3. Page 16, paragraph 1st: “We created 2 models: the first was unadjusted, the second was adjusted for age, sex, cigarette smoking, use of lipid lowering medication (all exposure groups), abdominal circumference (all exposure groups except physical inactivity)”. Abdominal circumference was an important association among lean individuals that developed NAFLD during follow up. Not sure why physical activity was excluded in this analysis, perhaps related to lack of association in lean individuals.

4. Please provide a flow sheet for exclusions for better presentation

5. Page 24, paragraph 2nd: “In univariate analysis, elevated BP was associated with greater than 2-fold risk in NAFLD which remained statistically significant even after controlling for likely confounders (IRR 1.40 [95%: 1.13 – 1.73])”. In table 3, elevated blood pressure lost statistical significance after adjusting for cofounders 1.19 (0.96-1.48)

6. Page 25, paragraph 1st: “However, in nonlean individuals, decreased physical activity was associated with incident NAFLD in both univariate and multivariate analysis (adjusted IRR 1.32 [ 95% CI: 1.14 – 1.54]). See table 5”. I am guessing the authors are referring to table 4 in this paragraph.

7. Page 28, paragraph 2nd: “The present study, which has significantly higher sample size (over 3000 lean participants), shows that in addition to elevated blood glucose, elevated BP and the presence of atherogenic dyslipidemia were independent risk factors for incident NAFLD among lean individuals”. According to table 4, elevated blood glucose was not associated with NAFLD among lean individuals in the adjusted model (1.45 (0.75 –2.79).

6. PLOS authors have the option to publish the peer review history of their article (what does this mean?). If published, this will include your full peer review and any attached files.

Reviewer #1: No

---

## [Author Response · Author response to Decision Letter 0]

1 Feb 2022

Response To Reviewers

PONE-D-21-09198R1 Cardiometabolic Disorders, Inflammation and the Incidence of Non-Alcoholic Fatty Liver Disease: A Longitudinal Study Comparing Lean and Non-lean Individuals

Editor’s Comments:

-What is the rationale for choosing 150 METs/min /week?

-METs and minutes per week are used interchangeably. Since they are not the same, please clarify what has been used. Is it 150 minutes/week of PA or 150 METs per week or 150METs/min/week. This need clarification.

We thank the Editor for their comments. The correct measure is minutes/week of physical activity and not METs/min/week. 150 minutes/week was chosen because of the guideline recommendations of physical activity by several medical societies including the American Heart Association. We have corrected this error where applicable throughout the manuscript so that the text now reads as “minutes/ week” instead of “150METs/min/week”. We apologize for the error and subsequent confusion. 

 -Why the author excluded participants with a BMI under 18 (kg/m2). Is there a rationale for NAFLD?

We excluded individuals with BMI <18.5kg/m2¬. Persons with BMI below 18.5 are considered underweight. Underweight persons are more likely to have other chronic illnesses, including malignancies and other metabolic conditions that may introduce unmeasurable confounders. In addition, inclusion of underweight individuals would not be consistent with the definition of lean NAFLD – NAFLD in individuals with normal BMI. We have added a statement to this effect in the methods.

-Did the author kept only people with Data for NAFLD at baseline AND follow up? This needs to be clarified as well as how many people lost for these variables.

We thank the Editor for their question. The data selection is noted in the results section. Only individuals without NAFLD at baseline were included in this study. We have made this clearer by including a figure that details the participant flow and exclusion criteria and participant sizes for the study.

I encourage the author to look at the checklist of the STROBE and address each point. For example, inclusion criteria and hypothesis should be in the manuscript or more precise. This is important since PLOS One Criteria require each study to be reported according to the appropriate guidelines.

Again, we appreciate the insights shared by the Editor. We have made edits throughout the manuscript to align the manuscript with the STROBE criteria. We have also completed the STROBE checklist and included it in the Supplementary material.

Reviewer #1: 

The current study evaluates cardiometabolic risk factors among lean and non-lean individuals who were followed for the development of NAFLD. The study showed some significant differences between cardiometabolic profile of lean and non-lean individuals that developed NAFLD during follow up. Few comments:

1.It seems like participants were followed every 6months to one year. It is not clear why patients underwent hepatic USG at every visit? Was it based on clinical indication or specific criterias or all the participants underwent as part of screening protocol?

All participants underwent as hepatic ultrasound as a part of screening protocol. They did not need to meet other prespecified criteria.

2. Abstract: “In lean individuals, AD, elevated blood glucose and elevated BP were significantly associated with NAFLD”. Table 4 should elevated glucose was not associated with NAFLD among lean individuals in adjusted model whereas elevated BP and AD were.

We thank the reviewer for their comment. We have now clarified this sentence in the abstract. “In lean individuals, AD, elevated blood glucose and elevated BP were significantly associated with NAFLD although for elevated blood glucose, statistical significance was lost after adjusting for possible confounders.”

3. Page 16, paragraph 1st: “We created 2 models: the first was unadjusted, the second was adjusted for age, sex, cigarette smoking, use of lipid lowering medication (all exposure groups), abdominal circumference (all exposure groups except physical inactivity)”. Abdominal circumference was an important association among lean individuals that developed NAFLD during follow up. Not sure why physical activity was excluded in this analysis, perhaps related to lack of association in lean individuals.

We thank the reviewer for their comment. Our intent was to explain that waist circumference was excluded from our regression analysis only when Physical Activity was analyzed as the exposure. Physical Activity is known to prevent NAFLD and also it is known to reduce abdominal circumference. It is biologically plausible that abdominal circumference may be an intermediate in the pathway associating physical activity to NAFLD and adjusting for abdominal circumference in this exposure population is likely to lead to overadjustment. 

4. Please provide a flow sheet for exclusions for better presentation

We thank the reviewer for the suggestion. We have included a flowchart detailing the inclusion/ exclusion criteria for this study in the supplementary material. 

5. Page 24, paragraph 2nd: “In univariate analysis, elevated BP was associated with greater than 2-fold risk in NAFLD which remained statistically significant even after controlling for likely confounders (IRR 1.40 [95%: 1.13 – 1.73])”. In table 3, elevated blood pressure lost statistical significance after adjusting for cofounders 1.19 (0.96-1.48)

We thank the reviewers for this observation. We have corrected this typographical error. The first sentence in the results subsection on Blood pressure now reads as: 

“In univariate analysis, elevated BP was associated with greater than 2-fold risk in NAFLD but lost its significance after controlling for likely confounders (IRR: 1.19 [95%: 0.96 – 1.48])”.

6. Page 25, paragraph 1st: “However, in non-lean individuals, decreased physical activity was associated with incident NAFLD in both univariate and multivariate analysis (adjusted IRR 1.32 [ 95% CI: 1.14 – 1.54]). See table 5”. I am guessing the authors are referring to table 4 in this paragraph.

We thank the reviewer for the observation. This is correct and we apologize for the error. We have corrected the appropriate text in the manuscript.

The last sentence in the paragraph on page 19 now reads as:

“However, in non-lean individuals, decreased physical activity was associated with incident NAFLD in both univariate and multivariate analysis (adjusted IRR 1.32 [ 95% CI: 1.14 – 1.54]) (See table 4).”.

7. Page 28, paragraph 2nd: “The present study, which has significantly higher sample size (over 3000 lean participants), shows that in addition to elevated blood glucose, elevated BP and the presence of atherogenic dyslipidemia were independent risk factors for incident NAFLD among lean individuals”. According to table 4, elevated blood glucose was not associated with NAFLD among lean individuals in the adjusted model (1.45 (0.75 –2.79). 

We thank the reviewer for their insightful comment. We have corrected this and it now reads “The present study, which has significantly higher sample size (over 3000 lean participants), shows that elevated BP and the presence of atherogenic dyslipidemia were independent risk factors for incident NAFLD among lean individuals. Blood glucose was not an independent risk factor for incident NAFLD in lean individuals.”

---

## [Editor Report · Decision Letter 1]

23 Mar 2022

Cardiometabolic Disorders, Inflammation and the Incidence of Non-Alcoholic Fatty Liver Disease: A Longitudinal Study Comparing Lean and Non-lean Individuals

PONE-D-21-09198R1

Dear Dr. Aneni,

We’re pleased to inform you that your manuscript has been judged scientifically suitable for publication and will be formally accepted for publication once it meets all outstanding technical requirements.

Kind regards,

Martin Senechal, PhD

Academic Editor

PLOS ONE
---

## [Editor Report · Acceptance letter]

28 Mar 2022

PONE-D-21-09198R1 

Cardiometabolic Disorders, Inflammation and the Incidence of Non-Alcoholic Fatty Liver Disease: A longitudinal study comparing lean and non-lean Individuals 

Dear Dr. Aneni:

I'm pleased to inform you that your manuscript has been deemed suitable for publication in PLOS ONE. Congratulations! Your manuscript is now with our production department. 

Kind regards, 

on behalf of

Dr. Martin Senechal 

Academic Editor

PLOS ONE